# Synthesis of Temperature/pH Dual-Responsive Double-Crosslinked Hydrogel on Medical Titanium Alloy Surface

**DOI:** 10.3390/gels11060443

**Published:** 2025-06-09

**Authors:** Yutong Li, Jiaqi Wang, Shouxin Liu

**Affiliations:** Key Laboratory of Applied Surface and Colloid Chemistry, Ministry of Education, School of Chemistry and Chemical Engineering, Shaanxi Normal University, Xi’an 710119, China; lyt15045546662@163.com (Y.L.); wjqsxdtdx@163.com (J.W.)

**Keywords:** hydrogel, medical titanium alloy, double crosslinking, sustained drug release, temperature/pH responsiveness

## Abstract

Medical titanium alloy Ti-6Al-4V (TC4) is widely used as a surgical implant material in biomedical fields owing to its superior biocompatibility, corrosion resistance, and mechanical performance, particularly for osseous integration applications. However, long-term contact of medical titanium-based implants with human soft tissues may induce infection and inflammation. To address these limitations, a drug-loading gel was designed to be synthesized on a TC4 surface to improve biointegration. Considering the critical regulatory roles of temperature and pH in physiological environments, this study synthesized a dual-responsive hydrogel using the temperature-sensitive monomers 2-(2-methoxyethoxy)ethyl methacrylate (MEO_2_MA) and oligoethylene glycol methacrylate (OEGMA) and the pH-sensitive monomer diethylaminoethyl methacrylate (DEAEMA), employing stereocomplexed polylactic acid as a physical crosslinker and *N*,*N*′-methylenebisacrylamide (MBA) as a chemical crosslinker. A polydopamine-based initiator was synthesized via dopamine functionalization with 2-bromoisobutyryl bromide (BIBB). The amphiphilic co-network hydrogel was grafted onto a modified TC4 surface through atom transfer radical polymerization (ATRP). Integration of the drug-loading gel and TC4 gives the implant an “active therapeutic” function by localized drug release. The results demonstrated that the energy storage modulus of the double-crosslinked gel matched that of human soft tissues. The gels exhibited efficient drug release.

## 1. Introduction

Titanium alloys were initially introduced to the biomedical field in the 1940s [1]. Following decades of development, titanium-based materials have become widely adopted for artificial bone implantation due to their superior biocompatibility, corrosion resistance, and mechanical performance [2,3]. Hijazi K et al. used titanium alloy implants for mandibular reconstruction [4]. Wang et al. used titanium alloy implants for bone defect treatment [5]. However, the storage modulus of medical-grade titanium alloys (*E* = 105–120 GPa) significantly exceeds that of human soft tissue (*E* = 1–100 kPa), creating mechanical mismatch challenges at the implant–tissue interface [6]. The modulus discrepancy often results in implant loosening Furthermore, peri-implant infections and inflammatory responses remain predominant contributors to implant failure, as evidenced by clinical studies [7,8,9]. Although systemic preoperative antibiotic prophylaxis has been proposed to mitigate postoperative infection, this approach is limited by systemic distribution issues and insufficient localized therapeutic efficacy [10]. To address these limitations, inspired by the structure and lubrication system of natural joints, researchers have tried to combine hydrogel with artificial bone to form a gradient composite material with a “hard base and soft surface” [11,12]. A critical challenge is achieving robust interfacial adhesion between hydrogel coatings and metallic implants. Surface modification of medical titanium alloys has been demonstrated to enhance gel–substrate bonding, offering a viable pathway to stabilize hydrogel-based therapeutic interfaces [13].

Surface modification techniques commonly employed in biomaterials engineering include anodization [14,15], sol-gel processing [16,17], plasma spraying [18,19], and ATRP [20,21]. Among these, ATRP has been predominantly adopted for synthesizing functional gel coatings due to its operational simplicity, mild reaction conditions, and broad monomer compatibility [22]. Stimuli-responsive hydrogels, characterized by their dynamic three-dimensional networks that adapt to environmental changes, have emerged as effective localized drug delivery platforms [23]. They are broadly categorized into temperature-responsive hydrogels [24], pH-responsive hydrogels [25], redox-responsive hydrogels [26], and dual-responsive hydrogels [27] according to the responsiveness. Dual-responsive hydrogels can respond to two different environmental stimuli, avoiding unexpected responses caused by a single stimulus and improving the reliability of the material. Depending on the crosslinking method, they can be categorized as physical crosslinked hydrogels [28,29] and chemical crosslinked hydrogels [30,31]. In recent years, the continuous optimization of the structure of stimuli-responsive hydrogels and the construction of multiple response have made them more promising in the field of biomedicine [32,33]. A representative application is demonstrated in the effective treatment of corneal abrasions through multistage drug delivery [34]. Synthesis of a drug-loading gel on a titanium alloy surface avoids direct contact between the titanium alloy and human soft tissues, makes up for the shortcomings of titanium alloy as an implant material such as large friction coefficient and poor cushioning ability. Localized sustained release of drugs improves the therapeutic effect of the implant. Collectively, these advancements demonstrate potential for reducing postoperative inflammatory responses while improving implant biointegration.

Inspired by mussel adhesion proteins, dopamine can form a polydopamine (PDA) film with robust adhesion and biocompatibility on diverse material surfaces [35]. Building on this strategy, researchers use the modified polydopamine coating as an initiator to synthesize hydrogels on the substrate surface [36,37]. This approach offers significant advantages, as both initiator immobilization and interfacial adhesion are achieved in a single reaction step.

Polylactic acid (PLA), a hydrophobic biopolymer synthesized from renewable resources, exhibits superior biocompatibility and degradability [38]. PLA exists in enantiomeric forms: poly(L-lactic acid) (PLLA) and poly(D-lactic acid) (PDLA) [39]. Stereocomplexation between these enantiomers enables physical crosslinking. However, single physically crosslinked gels demonstrate structural instability and compromised mechanical integrity [40]. In order to improve the properties of the gels, chemical crosslinking was introduced. The temperature-sensitive monomers 2-(2-methoxyethoxy)ethyl methacrylate (MEO_2_MA) and oligoethylene glycol methacrylate (OEGMA) as polyethylene glycol (PEG) analogs are intrinsically hydrophilic and biocompatible [41]. The lower critical solution temperature (LCST) of the polymers can be adjusted by changing the ratio of the monomers, making the LCST close to 37 °C [23,42]. The pH-sensitive monomer diethylaminoethyl methacrylate (DEAEMA) contains tertiary amine groups that undergo reversible protonation or deprotonation around its pKa [43], which imbues the gel with pH sensitivity.

Given the critical roles of temperature and pH in physiological systems, a physicochemical dual-crosslinked amphiphilic co-network hydrogel was designed on medical titanium alloy surfaces to enhance bio-integration. This system was synthesized via surface-initiated atom transfer radical polymerization (SI-ATRP) using MEO_2_MA, OEGMA and DEAEMA monomers, stereocomplexed PLA as a physical crosslinker, *N*,*N*′-methylenebisacrylamide (MBA) as a chemical crosslinker, and functionalized PDA as the initiator.

## 2. Results and Discussion

### 2.1. Synthesis of Hydrogels

HEMA-PLLA_20_ and HEMA-PDLA_20_ were synthesized via DBU-catalyzed ring-opening polymerization of hydroxyethyl methacrylate (HEMA) with L-lactide and D-lactide, respectively. These macromolecular monomers were ultrasonically treated and stereocomplexed to form the physical crosslinker (HEMA-scPLA_20_), as illustrated in Figure 1. Physicochemical dual-crosslinked hydrogels were subsequently fabricated on modified TC4 surfaces using monomers OEGMA, MEO_2_MA and DEAEMA, with MBA as the chemical crosslinker and HEMA-scPLA_20_ as the physical crosslinker. The synthetic procedure is detailed in Figure 2. Hydrogels were synthesized by varying the content of physical crosslinker to comparatively study the properties of hydrogels.

Hydrogels were synthesized by varying the physical crosslinker content. Table 1 shows the synthesis data of the double-crosslinked hydrogels.

### 2.2. Structural Characterization

#### 2.2.1. Characterization of HEMA-PLLA_20_

Figure 1a shows the ^1^H NMR spectrum of HEMA-PLLA_20_. From the spectrum, it can be concluded that 6.10 ppm (a) and 5.58 ppm (b) are vinyl protons peaks, 5.10–5.26 ppm (c) and 1.41–1.61 ppm (f) are proton peaks of -(C=O)-CH(CH_3_)-submethylene and methylene, 4.29–4.39 ppm (d) are -OCOCH_2_CH_2_O- group sub methyl proton peaks, and 1.92 ppm (e) is the methyl proton peak of -(CH_3_)CCH_2_ group. The ^1^H NMR spectrum of HEMA-PDLA_20_ is consistent with HEMA-PLLA_20_. ^1^H NMR spectra proved the successful synthesis of the macromolecular monomers.

The number of LA units in HEMA-PLLAn and HEMA-PDLAn is calculated using the following formula:$$n=\frac{A_{c}}{A_{a}}$$
where *Ac* and *Aa* represent the integrated area of peak c and peak a, which are mentioned in Figure 1. HEMA-PLLA_20_, *Aa* = 1.00, *Ac* = 20.39, n = 20.39 ≈ 20.

#### 2.2.2. XRD Characterization of HEMA-scPLA_20_

The physical crosslinker HEMA-scPLA_20_ was obtained through ultrasonic stereocomplexation of HEMA-PDLA_20_ and HEMA-PLLA_20_. Figure 1b shows the XRD spectra of HEMA-scPLA_20_ and HEMA-PLLA_20_. In the spectrum of HEMA-PLLA_20_, the characteristic peak of the macromolecular monomer appeared at 2*θ* = 17°, which proved the successful synthesis of the HEMA-PLLA_20_. Comparative analysis revealed disappearance of this 17° peak in HEMA-scPLA_20_, with new diffraction peaks emerging at 2*θ* = 12°, 21°, and 24°. These peaks are assigned to stereocomplex crystallinity between PLLA and PDLA, confirming both successful HEMA-scPLA_20_ synthesis and the formation of a physical crosslinker via non-covalent interactions.

#### 2.2.3. Characterization of the TC4-Initiator

Figure 2a shows the FT-IR spectra of TC4, hydroxylated titanium alloys TC4-OH and TC4-initiator. In Figure 2a, broad and strong hydroxyl peaks appeared at 2900–3640 cm^−1^, indicating TC4-OH had been synthesized successfully. The spectrum of the TC4-initiator showed two weak absorption peaks at 1705 cm^−1^ and 1635 cm^−1^. The weak absorption peak at 1705 cm^−1^ represented the ester carbonyl (C=O) formed by the reaction of the hydroxyl group in PDA with BIBB [22]. The 1635 cm^−1^ absorption is attributed to amide carbonyl (C=O) stretching vibrations resulting from PDA amine group interactions with BIBB [44]. Collectively, The appearance of the above characteristic peaks indicates the successful synthesis of modified polydopamine initiator on TC4.

In order to visualize the two weak absorption peaks at 1705 cm^−1^ and 1635 cm^−1^ of the TC4-initiator’s spectrum more intuitively, we enlarged the FT-IR spectra in the wavelength range of 1400–2000 cm^−1^, and Figure 2b shows the locally enlarged FT-IR spectra.

Figure 3 shows the surface view of TC4-OH (a) and TC4-initiator (b). Macroscopically, the successful grafting of BIBB caused the surface of the titanium alloy to change from silver-white to yellow compared with TC4-OH. Therefore, the successful synthesis of the TC4-initiator can also be initially judged from appearance.

#### 2.2.4. XPS Characterization of TC4-Initiator

X-ray photoelectron spectroscopy (XPS) was performed to analyze the chemical elements of the titanium alloy surface. Figure 4a shows the XPS spectra of bare-TC4 and TC4-initiator. (b–f) are the spectra of the detected elements. The spectrograms were analyzed, the C 1s peak (b) and O 1s peak (d) were detected on both bare-TC4 and TC4-initiator surfaces. Compared with the TC4-initiator, bare-TC4 has a distinct Ti 2p peak (e) with a peak value of 457.66 eV. In the spectra of TC4-initiator, the Ti 2p peak disappears and exhibits an enhanced N 1s peak (c) along with the appearance of a Br 3d peak (f). The XPS results proved the successful synthesis of the TC4-initiator.

#### 2.2.5. Structural Characterization of Gels

Figure 5 shows the FT-IR spectra of gel1, gel2, and gel3. Since the compositions of the gels are same, the FT-IR spectra of the three gels are basically the same as a whole. The absorption peak at 1120 cm^−1^ is the C-O-C stretching vibration. The absorption peak at 1728 cm^−1^ is the C=O stretching vibration. The absorption peak at 2910 cm^−1^ is the bending and stretching vibration of methyl and methylene C-H. The appearance of the above absorption peaks can prove the successful synthesis of gels.

### 2.3. Morphological Analysis of Gels

Hydrogels possess three-dimensional network structure that provides the structural basis for drug loading. Figure 6 shows SEM images of gel1 (a), gel2 (b), and gel3 (c) at identical magnification. It can be seen that all gels have a porous structure. As the content of physical crosslinker increases, the number of interaction points in the hydrogel increases, the crosslinking density increases, and the pore size of the gel decreases.

### 2.4. Bonding Between the Hydrogel and TC4

Gels were immersed in secondary distilled water to achieve swelling equilibrium at 25 °C for 12 h. Due to the three-dimensional structure of the hydrogel, inter-facial stresses are generated with the substrate surface during the swelling process, resulting in poor bonding [45]. If the bonding between the hydrogel and TC4 surface is weak, the hydrogel will fall off from the substrate surface after swelling. On the contrary, if the bonding is strong enough, the hydrogel can still adhere to the substrate surface after swelling, as shown in Figure 7a. To further examine the bonding ability between the hydrogel and TC4, we synthesized a certain thickness of gel between two pieces of TC4, as shown in Figure 7b. We tried to separate the above two pieces of TC4 with tweezers, as shown in Figure 7c. It can be seen that stretching of the interlayer gel occurred during the separation process. After swelling, separating the above two pieces of TC4, it was found that the gel was fractured. The fracture was not from the grafting site between the gel and TC4, but internal gel, as shown in Figure 7d. This proves that there is a strong adhesion between the hydrogel and TC4.

### 2.5. Temperature Sensitivity of Gels

The equilibrium swelling rate of the hydrogel at different temperatures was tested by placing gel1 in secondary distilled water at pH = 7 and in the temperature interval of 34–40 °C for 12 h to achieve equilibrium swelling. Figure 8a shows the curve of the equilibrium swelling rate of hydrogel at different temperatures. It can be seen that the swelling rate of the gel showed a significant decrease at 36–37 °C. The LCST of the gel was controlled at 36–37 °C, which is close to the normal temperature of the human body, by adjusting the ratio of the temperature-sensitive monomers MEO_2_MA and OEGMA. Since gel1, gel2, and gel3 have the same proportion of temperature-sensitive monomers, the LCST of the gels was examined by taking gel1 as an example only. The change in the swelling rate with temperature also proves that the gels are temperature-sensitive.

To further verify the temperature sensitivity of hydrogels, gels were immersed in secondary distilled water at 37 °C to test the swelling rate. As shown in Figure 8b, all of gels exhibited swelling behavior. The swelling rate of hydrogel reached the maximum value when the immersion time was 3.5 h. After 3.5 h, the swelling rate of the gels showed a decreasing trend. This is because the ambient temperature of 37 °C is higher than the LCST of the gels. The polymer chains in the hydrogels will realize the transition from hydrophilic to hydrophobic, resulting in a decrease in the swelling rate [46]. This indicates that hydrogels are temperature-sensitive. Specifically, the higher the content of physical crosslinker, the greater the number of interaction points within the hydrogel network. The crosslinking density was directly proportional to the content of the crosslinking agent. The hydrogel with the lowest crosslinking density had the highest swelling rate. Gel1, which possessed the lowest physical crosslinker content, demonstrated the highest equilibrium swelling rate, a result consistent with the SEM analysis.

### 2.6. pH Sensitivity of Gels

PBS buffer solutions were prepared with different pH, respectively, pH = 5.3, pH = 7.3, and pH = 9.3. The three gels were immersed in the above solution at *T* = 37 °C for 12 h to reach the equilibrium of swelling. Figure 8c shows the swelling of the gels at different pH, and all of gels have swelling behavior.

Under identical pH conditions, gel1 possessed the lowest physical crosslinker content and crosslinking density, resulting in the highest swelling ratio. Varying pH conditions, the swelling rate of the gel displayed regular variations with pH. The pH sensitivity of the gel was primarily attributed to the monomer DEAEMA (p*K*a = 7). When the pH was less than 7, the tertiary amino group of DEAEMA became protonated, generating an electrostatic repulsion effect within the gel. This caused an increase in pore size, enabling the gel to absorb water and swell. Conversely, when the ambient pH was greater than 7, the tertiary amino group of DEAEMA underwent deprotonation, eliminating the electrostatic repulsion effect. As a result, the gel lost water, leading to a decrease in the swelling rate. Therefore, gel1 demonstrated the highest swelling at pH = 5.3 and the lowest swelling at pH = 9.3. Similarly, gel2 and gel3 exhibited swelling behaviors consistent with gel1.

### 2.7. Amphiphilicity of Gels

The gels synthesized in this study were co-network hydrogels (APCNs), consisting of hydrophilic and hydrophobic chains interspersed with amphiphilic properties. The temperature-sensitive monomers (MEO_2_MA, OEGMA) and pH-sensitive monomer (DEAEMA) are hydrophilic. The macromolecular monomer (HEMA-PLA) is hydrophobic. To confirm the amphiphilicity of the gel, freeze-dried gel (a) was immersed in secondary distilled water at room temperature until swelling equilibrium, and photographed with a digital camera (c). Figure 9 shows digital photographs of the gel after complete swelling in the organic and aqueous phases, respectively. The gel was dried after swelling in water. The dried gel was immersed in the organic phase THF until complete swelling, and then photographed and recorded again (b). It can be seen that the gel can swell in both the organic and aqueous phases. The volume of swelling in the organic phase is smaller than that in the aqueous phase, which can prove that the synthesized gel has amphiphilicity.

### 2.8. Thermal Stability of Gels

Figure 10a shows the TG curves (thermogravimetric curves) of gel1, gel2, and gel3. The first weight loss temperature of the gel can be used to measure the thermal stability. From the TG curves, it can be seen that all of the gels lost weight only once around 300 °C, which can prove that the double crosslinked gels have good thermal stability.

### 2.9. Mechanical Properties of Gels

The synthesis of gel on medical titanium alloys is aimed at addressing mechanical compatibility issues arising from long-term contact with human soft tissues. Therefore, the energy storage modulus of the TC4 surface gel needs to match with that of human soft tissue (1–100 kPa).

The three gels were immersed in secondary distilled water until swelling equilibrium at 37 °C. Figure 10b shows the curves of the energy storage modulus of the three gels. At a frequency of 10 Hz, the energy storage modulus of gel1, gel2, and gel3 were 12.76 kPa, 18.42 kPa, and 19.78 kPa, which all matched with those of human soft tissues. The content of the physical crosslinking agent affects the network structure of the gels. Gel3 has the highest content of physical crosslinker, so gel3 has the tightest three-dimensional network structure, the smallest pore size, the largest energy storage modulus, and the best mechanical properties.

Figure 10c shows the loss tangent (tan δ) of all gels at 10 Hz. Gel1, gel2, and gel3 exhibited loss tangents of 0.054, 0.066, and 0.051. The elasticity of the material is inversely proportional to the loss tangent. The loss tangent of the gels is small, which indicates that the three types of gels are predominantly elastic under the action of external forces.

### 2.10. Sustained Drug Release of Gels

Poor biointegration of the implant with soft tissues often triggers an inflammatory response at the implant site, reducing local pH(~5.3) below the normal physiological levels. Indomethacin, a widely used nonsteroidal anti-inflammatory drug (NSAID) in clinical practice, exhibits potent anti-inflammatory and antipyretic effects [47]. Consequently, we synthesized a drug-loading gel using indomethacin as the model drug. The drug release behavior of the gel was evaluated by immersing the drug-loaded gel in PBS buffer solutions at *T* = 37 °C and pH = 5.3 to simulate inflammatory micro-environment. Additionally, drug release at *T* = 37 °C and pH = 7.3 was investigated as a control group to assess the pH-responsive targeting capability of the drug release system.

The absorbance of Indomethacin solution (solvent is PBS buffer of pH = 7.3) was tested in the wavelength range of 200–600 nm using a UV-visible spectrophotometer with PBS buffer solution of pH = 7.3 as a blank control. The maximum absorption wavelength was measured to be 320 nm.

Figure 11 shows the standard curve of Indomethacin solution. The absorbance of different concentrations of Indomethacin solution was tested at the maximum absorption wavelength of 320 nm, using PBS buffer solution at pH = 7.3 as solvent. The standard curve was plotted using the concentration of Indomethacin solution as the horizontal coordinate and absorbance as the vertical coordinate.

All gels were placed in PBS buffer at *T* = 37 °C, pH = 5.3 to simulate the physiological parameters of the inflammation site. Figure 12a shows the cumulative drug release rates of the gels. After 72 h, the drug release rates of gel1, gel2, and gel3 were 68.89%, 61.11%, and 48.73%. The drug release rates of the gels were high and showed regular changes. The drug release rate of gels was inversely proportional to the content of physical crosslinker.

In order to verify the targeting of drug release, gel1 was placed in environment of *T* = 37 °C and pH = 7.3 to simulate normal physiological parameters of the human body. Figure 12b shows the drug release rate under different pH conditions. Under the condition of pH = 7.3, the drug release rate of gel1 was 62.29% < 68.89%. The drug release rate at the site of inflammation was higher than that in a normal physiological environment. The above results showed that the pH response performance of the gel was sufficient to achieve the controlled release of the drug and had certain targeting properties. The combination of drug-loading gel and medical implant enhances the therapeutic effect of implant through local drug sustained release, which is expected to reduce postoperative inflammation. In this part, the experiments were conducted with a sample size of one, so there will be some deviations compared with the values determined by parallel experiments.

## 3. Conclusions

In this study, physicochemical dual-crosslinked amphiphilic co-network drug-loading hydrogels were successfully synthesized on medical titanium alloy surface via ATRP. A modified polydopamine initiator was employed, with poly(lactic acid) stereocomplexation severing as the physical crosslinker and MBA as the chemical crosslinker. Temperature-sensitive monomers (MEO_2_MA, OEGMA) and a pH-sensitive monomer (DEAEMA) with good biocompatibility were introduced to make the gels temperature-/pH-sensitive. The test results showed that the synthesized dual-responsive double-crosslinked hydrogels’ LCST was successfully controlled at 36–37 °C, which was close to human body temperature and had good stability. At 37 °C, the energy storage modulus of the hydrogel matched that of human soft tissues, which was favorable for the biointegration of medical implants. Meanwhile, the porous network enabled drug loading and the drug sustained release results of the gel showed that the cumulative drug release rate of the synthesized hydrogel reached 68.89% under the simulated inflammatory microenvironment. The drug release rate was high and the drug had certain targeting. Compared with existing studies [20], the dual crosslinked drug-loaded gel synthesized on the surface of TC4 can respond to the two critical physiological parameters of the human body. Through the sustained release of local drugs, and it endows the implant with “active therapeutic” function, which is expected to solve problem such as postoperative infection of traditional implants. Future in vivo studies will be required to confirm biocompatibility and infection prevention efficacy, which has broad application prospects in the biomedical field.

## 4. Materials and Methods

### 4.1. Materials

Titanium alloy (TC4) was purchased from Shaanxi Baotai Group Co. (Xi’an, China.) Hydroxyethyl methacrylate (HEMA, 99%) and 1,8-diazabicyclo[5.4.0]undec-7-ene (DBU, 98%) were purchased from Beijing Bailing Wei Technology Co. (Beijing, China) L-propylacrylate (L-LA, 98%), 2-(2-methoxyethoxy)ethyl methacrylate (MEO_2_MA, 97%), oligoet-hylene glycol methacrylate (OEGMA, 95%), and diethylaminoethyl methacrylate (DEAEMA, 99%) were purchased from Shanghai McLean Biochemistry and Technology Co. (Shanghai, China) D-propyllactone (D-LA, 99%) was purchased from Aladdin Chemical Reagent Co. (Shanghai, China) 2-Bromoisobutyl bromide (BIBB, 98%) and triethylamine (TEA, 99%) were purchased from Shanghai Sinopharm Chemical Reagent Co. (Shanghai, China). *N*,*N*,*N*′,*N*″,*N*″-Pentameth-yldiethylenetriamine (PMDETA, 98%) and *N*,*N*′-methylenebisacrylamide (MBA, 99%) were purchased from TCI (Shanghai Development Co., Ltd. (Shanghai, China)). Dopamine (DA, 95%) was purchased from Wuhan Jiangsu Akcome Biomedical Research and Development Co., Ltd., (Wuhan, China). *N*,*N*′-Dimethylformamide (DMF) ultra-dry solvent and tetrahydrofuran (THF) ultra-dry solvent were purchased from Anegi Chemicals Ltd. (Shanghai, China) Dichloromethane (DCM) was purified and the experimental water was secondary distilled water.

### 4.2. Methods

#### 4.2.1. Synthesis of HEMA-PDLA_20_

The whole reaction was carried out under N_2_ atmosphere. D-lactide (4.6 mmol), the initiator HEMA (0.46 mmol) and 25 mL purified DCM were added to a dry three-necked flask. After dissolution of all solids, the catalyst DBU (0.1 mL) was added to the flask. The mixture was stirred at room temperature for 12 h and quenched with benzoic acid (0.1 g). The resultant mixture was transferred to a beaker containing 30 mL of hexane under ice-water bath conditions. Solvent evaporation was performed in the fume hood. The resulting precipitate was the macromonomer HEMA-PDLA_20_. The synthesis of HEMA-PLLA_20_ was carried out in agreement with that of HEMA-PDLA_20_. and only the enantiomer of PLA was changed.

#### 4.2.2. Synthesis of HEMA-scPLA_20_

A total of 0.05 g HEMA-PLLA_20_ and 0.05 g HEMA-PDLA_20_ were placed in a 10 mL round bottom flask. In total, 3 mL THF was added to dissolve it. The mixture was sonicated for 1 h to fully stereocomplex at 25 °C. After stereocomplex, all solids were dissolved and the solution was translucent white. The flask was placed in a rotary evaporator at 45 °C for 10 min. The vacuum level was 0.07 MPa. The solvent was spun dry, and the resulting translucent white viscous solid was the physical crosslinker HEMA-scPLA_20_.

#### 4.2.3. Synthesis of TC4-Initiator

The size of TC4 was 1 cm × 1 cm. TC4 was ultrasonically cleaned with acetone, ethanol, and distilled water in turn after being polished. TC4 was soaked in each solvent and cleaned by ultrasound for 10 min, totaling 30 min. The cleaned TC4 was immersed in 6 mol/L NaOH solution at 65 °C for 24 h. Then, TC4 was repeatedly washed with distilled water and placed in a vacuum drying oven at 65 °C for 24 h to obtain TC4-OH.

Under an N_2_ atmosphere, dopamine (200 mg, 1.05 mmol) was added into three-necked flask and dissolved in *N*,*N*′-dimethylformamide (DMF, 10 mL). The reaction system was placed under an ice water bath (*T* = 0–4 °C). 2-bromoisobutyl bromide (BIBB, 0.065 mL, 0.525 mmol) and triethylamine (0.075 mL, 0.525 mmol) were added slowly and dropwise in turn. The ice water bath was removed after 5 min of reaction and the mixture was stirred at room temperature for 3 h. The mixture was then transferred to a beaker containing 50 mL 0.04 M tris buffer solution (pH = 8.5). TC4-OH was immersed into the beaker containing the above mixture and stirred for 18 h at room temperature. The above titanium sheet was washed with distilled water and dried under reduced pressure at 65 °C. TC4-initiator was obtained.

#### 4.2.4. Synthesis of Gels

As an example for gel3, a dry Shrek tube was filled with the monomers OEGMA (0.3 mmol), MEO_2_MA (2.7 mmol), DEAEMA (0.6 mmol), the physical crosslinker HEMA-scPLA_20_ (0.2 g), the chemical crosslinker MBA (1.2%), the ligand *N*,*N*,*N*′,*N*″,*N*″-Pentamethyldi ethylenetriamine (PMDETA) and solvents (0.07 mL MeOH and 0.07 mL H_2_O). MeOH and secondary water were added sequentially. The mixture was repeatedly frozen and thawed three times under N_2_ atmosphere to achieve an oxygen-free state. Freezing was achieved by liquid nitrogen. There are no bubbles inside the liquid and the liquid level beating represents the completion of deoxygenation. Then a small amount of catalyst cuprous bromide (CuBr, 1 mg) was quickly added. After 5 min, TC4-initiator was added. These operations must be rapid and minimize contact with the air. The closed Shrek tubes were placed in an oil bath overnight at 65 °C for polymerization and crosslinking. The synthesis steps for gel1 and gel2 were the same as for gel3.

### 4.3. Structural Characterization

#### 4.3.1. Proton Nuclear Magnetic Resonance Spectroscopy (^1^H NMR)

^1^H NMR spectrum of HEMA-PLLA_20_ was recorded on a 400 MHz 1H NMR spectrometer model JNM-ECZ400S/L1 from Shizuoka, Japan. Before measurement, the samples to be measured were dissolved with CDCl_3_.

#### 4.3.2. Fourier-Transform Infrared Spectroscopy (FT-IR)

Frontier ATR-FTIR from PerkinElmer was used to record spectrum of the unmodified TC4, TC4-OH, and TC4-initiator. The wavenumber range of the scan was 4000 cm^−1^–500 cm^−1^. Scanning was performed in SLR mode. The resolution of the instrument was less than 4 cm^−1^. The number of scans was 64. The samples were kept dry before measurement. All of FT-IR tests covered in this study were background subtracted and baseline corrected.

The gels were measured with an NIR-MIR spectrometer Tensor II from Bruker, Germany. Freeze-dried gels were immersed in absolute ethanol for 2 h. The soaked gel was ground with KBr. The mixture was placed in an oven at 70 °C to dry.

#### 4.3.3. X-Ray Photoelectron Spectroscopy (XPS)

Elemental changes on the surface of TC4 and TC4-initiator were recorded on Escalab Xi+ X-ray photoelectron spectroscopy from Beijing, China. The X-ray source was Al Kα. The pass Energy was 130.0 eV.

#### 4.3.4. X-Ray Diffraction (XRD)

A powder X-ray diffractometer with specifications of Bruker D8 Advance produced by Bruker Company, Karlsruhe, Germany, was used to record the diffraction peaks of the macromolecular monomers and physical crosslinker. The voltage was 40 kV and the electric current was 40 mA. The scanning range was 5–30° and the scanning speed was 2°/min. The samples were uniformly coated on the template for testing.

### 4.4. Temperature Sensitivity of Gels

Three freeze-dried gels were placed in secondary distilled water within the temperature range of 34–40°C and pH = 7 to achieve equilibrium swelling. The equilibrium swelling rates of the hydrogels at different temperatures were tested.

The freeze-dried gels were cut into pieces, weighed, recorded, and immersed in secondary distilled water at *T* = 37 °C, pH = 7. The samples were removed at regular intervals, weighed, and recorded. The swelling rate of gel can be calculated according to the following equation:*Swelling Ratio* = (*W_t_* − *W*_d_)/*W*_d_(1)
where *W_t_* is the mass (g) of the gel at the time of swelling to *t*, and *W*_d_ is the mass (g) of the dried gel.

### 4.5. pH Sensitivity of Gels

The freeze-dried gels were cut into pieces, weighed, recorded, and immersed in PBS buffer solution at *T* = 37 °C, pH = 5.3, pH = 7.3, and pH = 9.3 for 12 h to reach swelling equilibrium. After taking out, they were weighed and recorded again. The swelling rate of the hydrogels was calculated according to the above Equation (1).

### 4.6. Amphiphilicity of Gels

The freeze-dried gels were cut, weighed, recorded, and immersed in secondary distilled water at room temperature until equilibrium swelling. After taking out, they were photographed and recorded. The equilibrium swollen gel was dried, immersed in THF to equilibrium swelling, photographed, and recorded.

### 4.7. Scanning Electron Microscopy (SEM) Analysis of Gels

The morphology of the gels was analyzed using a scanning electron microscope. The equilibrated solubilized hydrogel was quickly frozen with liquid nitrogen and freeze-dried before testing.

### 4.8. Thermal Stability Analysis of Gels

A thermal analysis system (STA449F5) was used to determine the thermal stability of the hydrogel. The mass of the hydrogel sample was about 5 mg. The working conditions of the instrument were as follows: under dry N_2_ atmosphere, with a heating rate of 10 °C/min and a scanning range of 25–600 °C.

### 4.9. Mechanical Properties Testing of Gels

An Anton Paar MCR302 rheometer from Graz, Austria was used to determine the energy storage modulus and loss angle tangent of the hydrogels. The gels were placed in secondary distilled water at 37 °C until swelling equilibrium before testing.

### 4.10. Sustained Drug Release of Gels

Synthesis of drug-loading gel: 5 mg of Indomethacin was mixed with monomer and drug-loading gel was synthesized by ATRP. Indomethacin as a target drug was embedded in the pore structure of the gel through hydrophobic interactions, a behavior that is physical and does not affect the crosslinking behavior of the gel.

Determination of maximum absorption wavelength: Indomethacin solution was prepared at a concentration of 50 μg-mL^−1^. Its maximum absorption wavelength was measured at 320 nm by spectral scanning between 200 and 600 nm using the UV-visible spectrophotometer model UV-1901 with PBS buffer as the blank control group.

Determination of standard curve: PBS was used as the solvent to prepare Indomethacin solutions with different concentration gradients in the range of 1–50 μg-mL^−1^ (the concentrations were 1 μg-mL^−1^, 5 μg-mL^−1^, 10 μg-mL^−1^, 15 μg-mL^−1^, 20 μg-mL^−1^, 25 μg-mL^−1^, 50 μg-mL^−1^). And the absorbance at different concentrations was measured at the maximum absorption wavelength of 320 nm. The linear regression equation was fitted, with the horizontal coordinate as the concentration of Indomethacin and the vertical coordinate as the absorbance:*Abs* = 0.01958*c* + 0.00313*R*^2^ = 0.9952
where *Abs* is the absorbance of Indomethacin at the maximum absorption wavelength of 320 nm, and *c* is the concentration of Indomethacin

In vitro sustained release of the drug: The drug-loading gel was immersed in 150 mL of PBS buffer solution at *T* = 37 °C pH = 7.3 and pH = 5.3, respectively. In total, 4 mL of the sustained-release solution was taken out at certain intervals, whose absorbance was measured at the maximum wavelength. Then, an equal volume of fresh PBS buffer solution was added to the system. The amount of drug released at different times was obtained from the standard curve and finally the cumulative release rate of Indomethacin was calculated from the following equation:$$Cumulative\;release\;\left(\%\right)=\frac{V_{e}\sum_{1}^{n-1}C_{i}+{V_{0}C}_{n}}{m_{drug}}\times 100$$
where *Ve* is the volume of solution removed from the PBS buffer solution each time, *V*_0_ is the total volume of solution released, ***c_i_*** is the concentration of drug released from the gel, and *m_drug_* is the total mass of Indomethacin loaded into the gel.

## Data Availability

The original contributions presented in this study are included in the article. Further inquiries can be directed to the corresponding author.

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
