# Peer review of "Synthesis of Temperature/pH Dual-Responsive Double-Crosslinked Hydrogel on Medical Titanium Alloy Surface"

_gels, 2025, doi:10.3390/gels11060443_

Round 1

Reviewer 1 Report

Comments and Suggestions for Authors

This work reports a promising method for developing dual-responsive hydrogel coatings on medical titanium alloy surfaces. The combination of both temperature and pH stimuli to achieve sustained drug release is highly important and relevant to biomedical implant applications. However, substantial improvements are needed before the manuscript can be published. Followings are the reviewer’s comments:

  • Please provide more literature survey in the Introduction and discussion parts for a comprehensive discussion of recent and relevant works on responsive hydrogels used in biomedical applications such as drug delivery, wound healing, and implant coatings. The authors may enrich the manuscript by referencing these papers [https://doi.org/10.1016/j.cej.2021.132409; https://pubs.acs.org/doi/10.1021/acsapm.3c03066; https://doi.org/10.1016/j.mtbio.2021.100186; https://doi.org/10.1016/j.mtbio.2024.100998 ].
  • Many of the figures appear fragmented and disrupt the visual flow of the manuscript. Figures can be grouped and organized logically. For example, for material characterizations (Figures 1-6): These can be grouped into 1 or 2 composite figures; Swelling behavior under temperature and pH (Figures 9-11) can be arranged together in one figure; Mechanical and thermal properties (Figures 13-15) can be unified into a broader figure summarizing the mechanical and thermal stability.
  • Quantitative data across the manuscript (e.g., swelling ratios, drug release percentages) are presented without error bars or any statistical treatment. This is a critical issue for biomedical material studies. Please include.
  • Minor language polishing is needed for smoother transitions (for example: replacing “So in this paper...” with “Therefore, this study...”).

Author Response

Dear reviewer,

The reply has been written in the attachment. Thanks!

Reviewer 2 Report

Comments and Suggestions for Authors

Line 10–11:

"Medical titanium alloy Ti-6Al-4V (TC4) is widely used as a surgical implant material in biomedical fields, particularly for osseous integration applications."

Comment: This is a good opening sentence. However, consider briefly mentioning why TC4 is favored (e.g., biocompatibility, corrosion resistance, mechanical strength) to support the claim and enrich the context.

Lines 31–33:

"Titanium alloys were initially introduced to biomedical field in the 1940s... adopted for artificial bone implantation owing to their superior biocompatibility, corrosion resistance, and mechanical performance[2, 3]"

Comment: The historical context is appropriate. However, consider adding specific examples of current orthopedic or dental applications to better contextualize relevance in clinical practice.

ine 34–35:

"the storage modulus of medical-grade titanium alloys significantly exceeds that of human soft tissue (E=1–100 kPa), creating mechanical mismatch..."

Comment: This is a critical observation. Specify the modulus range for TC4 alloy to highlight the contrast. Consider also referencing potential consequences.

Lines 41–43:

"inspired by... natural joints, hydrogel as a drug carrier for topical therapy... combine hydrogel with artificial bone..."

Comment: The sentence construction is unclear and grammatically incorrect.

Lines 52–55:

"Stimuli-responsive hydrogels... categorized into... responsive hydrogels[25]"

Comment: Consider summarizing advantages of dual-responsive hydrogels in one line

Line 387:

"D-lactide (4.6 mmol), the initiator HEMA (0.46 mmol) and 25 mL DCM..."

 Comment: Add the molecular weights or purity levels of D-lactide and HEMA. Also, mention whether DCM was anhydrous or distilled, as moisture can affect ring-opening polymerization.

Line 389:

"the catalyst DBU (0.1 mL) was added."

Comment: Please include concentration or molar amount of DBU for reproducibility. Also, mention how it was introduced.

Line 390:

"stirred at room temperature for 12h..."

Comment: Duration and temperature are clear. Still, please clarify exact temperature if known (e.g., 25 ± 1 °C) and whether reaction progress was monitored.

Lines 390–391:

"quenched by the addition of benzoic acid. The resulting mixture was added to a beaker containing 30 mL of hexane under ice-water bath conditions..."

Comment: Indicate amount of benzoic acid used. Also, clarify whether the precipitation was carried out by slow addition of the reaction mixture into hexane, or vice versa. These details affect polymer recovery and purity.

Line 394:

"The synthesis of HEMA-PLLA20 was carried out in agreement with that of HEMA-PDLA20."

Comment: It's acceptable to reference the method, but specify the enantioisomer used (L-lactide) and whether any variations in yield, time, or purification steps were encountered.

Line 396–397:

“0.05 g HEMA-PLLA20 and 0.05 g HEMA-PLLA20 were placed in a 10 mL round bottom flask.”

Comment: There seems to be a typographical error. Did you mean “0.05 g HEMA-PLLA20 and 0.05 g HEMA-PDLA20”? Stereocomplexes require enantiomeric macromonomers.

Line 398:

“3 mL THF was added to dissolve it, and the mixture was sonicated for 1 h to fully stereocomplex.”

Comment: Provide details on:

  • Sonication power/frequency, and whether it was continuous or pulsed.
  • Temperature control during sonication, since elevated temperature can affect complexation.

Also clarify: was the solution visually clear before or after sonication?

Line 399:

“The flask was placed in a rotary evaporator and the solvent was spun dry...”

Comment: Add conditions for solvent removal:

  • Rotation speed (rpm),
  • Water bath temperature (°C),
  • Duration (minutes),
  • Vacuum level (if applicable).

Line 400:

“The resulting solid was the physical cross-linker HEMA-scPLA20.”

Comment: Indicate yield (%), appearance (color, texture), and any characterization data confirming stereocomplex formation (e.g., DSC, FTIR, XRD).

Line 402:

“The TC4 was ground and polished, and ultrasonically cleaned with acetone, ethanol, and distilled water for 10 min.”

Comment: Clarify:

  • Particle or sheet dimensions of TC4,
  • Polishing method (e.g., grit size),
  • Whether solvents were applied sequentially or as a mixture, and
  • Ultrasonication power/frequency.

Lines 403–404:

“...placed in 6 mol/L NaOH solution and treated at 65 °C for 24 h.”

Comment: This is standard surface hydroxylation for titanium alloys. Specify volume of NaOH per gram/sheet, and whether the vessel was sealed or open.

Also, duplicate drying steps are present in lines 404 and 405:

“The TC4 was then dried in a vacuum drying oven for 24 h.”
“The TC4-OH was then dried at 65 °C for 24 h.”

Please consolidate or clarify the sequence and label assignments (TC4 → TC4-OH).

Line 408:

“dopamine (200 mg, 1.05 mol)”

Comment: This seems like a typo — 200 mg of dopamine cannot equal 1.05 mol. Please correct this value. Likely intended: 1.05 mmol.

Lines 408–410:

“...dissolved in DMF (10 mL) and 2-bromoisobutyl bromide (BIBB)... and triethylamine... were added slowly and dropwise to the ice water bath.”

Comment:

  • Clarify whether BIBB and TEA were premixed or added separately.
  • Mention reaction vessel (e.g., 2-necked flask), and how long the addition took.
  • What was the temperature of the ice bath?

Lines 412–413:

“...transferred to a beaker containing tris buffer solution. TC4-OH was immersed into the beaker...”

Comment: Define:

  • pH and concentration of tris buffer,
  • Volume of buffer used,
  • Whether the reaction was under nitrogen.

Also clarify if any surface modification control experiments (e.g., dopamine-only) were done.

Line 418–419:

“As an example for gel3, a dry Shrek tube was filled with the monomers OEGMA (0.3 mmol), MEO2MA (2.7 mmol), and DEAEMA (0.6 mmol), the physical cross-linker HEMA-scPLA20, the chemical cross-linker MBA...”

Comment:

  • Clearly specify the amounts (mg or µL) used for HEMA-scPLA20 and MBA (N,N′-methylenebisacrylamide).
  • Mention molecular weights or purity of monomers, and supplier names, if possible.

Line 420:

“...the ligand N,N,N,N,N-Pentamethyldiethylenetriamine (PMDETA) and solvents (0.07 mL MeOH and 0.07 mL H₂O).”

 Comment:

  • Recheck and correct the ligand name: It should be N,N,N′,N′,N″-Pentamethyldiethylenetriamine (PMDETA), not “N,N,N,N,N.”
  • Clarify whether the solvents were added sequentially or as a premix, and whether they acted as co-solvents or to aid solubility of a specific component.

Line 422:

“The mixture was repeatedly frozen and thawed three times under N₂ atmosphere...”

Good step to eliminate oxygen for controlled radical polymerization.

Add:

  • Approximate duration of freezing and thawing cycles.
  • Temperature used for freezing (e.g., liquid nitrogen or −20 °C?).
  • If done manually or using a programmed freeze–thaw cycle.

Line 423–424:

“...a small amount of catalyst cuprous bromide (CuBr) was quickly added. After 5 min, TC4-initiator was added.”

Comment:

  • Specify the exact amount (mg or mol) of CuBr.
  • How was the initiator pre-treated or activated (if at all)?
  • What was the concentration of initiator sites on TC4?

Line 433:

“The unmodified TC4, TC4-OH, and TC4-initiator were measured using Frontier ATR-FTIR from PerkinElmer.”

Instrument and samples are well mentioned.

Add:

  • Was ATR-FTIR performed in single reflection or multiple reflection mode?
  • Wavenumber range scanned (e.g., 4000–500 cm⁻¹)?
  • Number of scans averaged and resolution (cm⁻¹).

Line 434:

“The gels were measured with a NIR-MIR spectrometer from Bruker, Germany.”

Comment:

  • Name the model of the Bruker instrument (e.g., Tensor II, Alpha II).
  • Describe sample preparation: Were the gels measured as films, disks, or powders?
  • If gels are soft, were KBr pellets or ATR used?

Line 418–419:
“As an example for gel3, a dry Shrek tube was filled with the monomers OEGMA (0.3 mmol), MEO2MA (2.7 mmol), and DEAEMA (0.6 mmol), the physical cross-linker HEMA-scPLA20, the chemical cross-linker MBA...”

Comment:
Clearly specify the amounts (mg or µL) used for HEMA-scPLA20 and MBA (N,N′-methylenebisacrylamide).

Line 423:
“Then a small amount of catalyst cuprous bromide (CuBr) was quickly added.”

Comment:
Specify the exact amount (mg or mmol) of CuBr added, and state the purity and supplier. Also clarify if it was added under inert conditions and how oxygen exposure was minimized.

Line 433:
“The unmodified TC4, TC4-OH, and TC4-initiator were measured using Frontier ATR-FTIR...”

Comment:
Include scanning parameters (e.g., spectral range, number of scans, resolution). Also, mention whether background subtraction and baseline correction were performed.

Line 434:
“The gels were measured with a NIR-MIR spectrometer...”

Comment:
Indicate whether samples were pressed into KBr pellets or measured directly as films/powders. Also mention the scanning mode (e.g., ATR, transmission).

Line 436–437:
“...measured chemical elements on the surface of bare-TC4 and TC4-initiator.”

Comment:
Mention operating conditions: X-ray source (e.g., Al Kα), pass energy, and analysis depth. Also, indicate whether charge neutralization was applied.

Line 441:
“Powder X-ray diffractometer... Bruker D8 Advance...”

Comment:
Provide scanning parameters: 2θ range, scan rate, step size, and voltage/current settings. Indicate whether the sample was pressed, coated, or free powder.

Line 475:
“5 mg of Indomethacin was mixed with monomer and drug-loading gel was synthesized by ATRP.”

 Comment:
Specify the monomer used and its ratio to Indomethacin. Mention whether drug incorporation affected gelation or cross-linking behavior.

Line 481–485:
“...absorbance at different concentrations was measured... linear regression equation was fitted.”

 Comment:
Mention how many concentrations were tested (n = ?), and whether the linearity was validated by residual analysis or %RSD.

Line 96:
“The above macromolecular monomers were ultrasonically and stereocomplexed to prepare the physical cross-linker (HEMA-scPLA20), as illustrated in Scheme 1.”

Comment:
Provide ultrasonic parameters (e.g., frequency, power, duration, temperature) used for stereocomplexation.

Lines 97–99:
“Monomers OEGMA, MEO2MA, DEAEMA, chemical cross-linker MBA, and physical cross-linker HEMA-scPLA20 were synthesized into physicochemical double-crosslinked hydrogels on modified TC4 surface. The synthesis steps are shown in Scheme 2.”

Comment:
Clarify whether polymerization was performed in solution or bulk, and detail initiator type, initiator concentration, and polymerization time/temperature.

Lines 122–127:
“The number of LA units in HEMA-PLLAn and HEMA-PDLAn is calculated using the formula n = Ac / Aa where Ac and Aa represent the integrated area of peak c and peak a, respectively. In Figure 1, HEMA-PLLA20, Aa = 1.00, Ac = 20.39, n = 20.39 ≈ 20.”

Comment:

  • Clearly define which specific protons correspond to peaks 'a' and 'c' in the polymer structure to avoid ambiguity (e.g., vinyl protons for 'a', methylene/methyl for 'c').

Justify the assumption that the integration of peak 'a' is normalized to 1.00—state if internal or external standards were used for calibration.

Lines 130–138:
“Figure 2(a) shows the FT-IR spectra of hydroxylated titanium alloys TC4-OH and TC4-initiator... The appearance of the above characteristic peaks proves the successful synthesis of modified polydopamine initiator on TC4.”

Comment:

Explain the rationale for assigning 1705 cm⁻¹ to ester carbonyl from PDA-BIBB reaction and 1635 cm⁻¹ to amide carbonyl—cite relevant literature or model compounds to support these assignments.

Lines 194–207:
“The synthesized gels were immersed in secondary distilled water to achieve swelling equilibrium... This shows that there is a strong adhesion between the hydrogel and TC4.”

Comment:

  • Clearly specify the swelling duration and conditions (e.g., temperature, time, agitation) to ensure consistency and reproducibility of the swelling equilibrium.
  • Consider adding control experiments with non-modified TC4 or hydrogels without the initiator to highlight the role of surface modification in bonding strength.

Lines 213–239:
“The equilibrium swelling rate of the hydrogel at different temperatures was tested by placing gel1 in secondary distilled water at pH=7 and in the temperature interval of 34 °C-40 °C to achieve equilibrium swelling...”

Comment:

  • Specify the exact method and duration used to determine the equilibrium swelling rate at each temperature. Include details such as immersion time at each temperature before measurement and whether the measurements were repeated for reproducibility.
  • Clarify how the swelling rate was calculated (e.g., formula or method for determining % swelling). State whether swelling was measured gravimetrically, volumetrically, or by another technique.
  • For Figure 10, specify the time intervals at which swelling measurements were taken during the 3.5 h period and afterward.

Lines 242–257:
“The three gels were immersed in PBS buffer solution at T=37 °C, pH=5.3, pH=7.3, and pH=9.3 for 12h to reach the equilibrium of swelling...”

Comment:

Clarify if the 12 h immersion time was confirmed as sufficient for equilibrium swelling at each pH, including whether time-dependent swelling data were collected to confirm plateau.

Suggest exploring swelling reversibility by cycling pH conditions to demonstrate the dynamic pH sensitivity of the hydrogels.

Lines 347–363:
“In this paper, physicochemical dual-crosslinked amphiphilic co-network drug-loading hydrogels were successfully synthesized on the surface of medical titanium alloys…”

Comment:

Overall summary is well-structured, clearly highlighting key outcomes of synthesis, responsiveness (pH and temperature), mechanical compatibility, and drug release performance.

Be more quantitative:

  • Include specific performance metrics where applicable (e.g., % drug release over time, LCST range, swelling ratio at different pH or temperatures, or energy storage modulus values).

Clarify novelty and impact:

  • Briefly emphasize what distinguishes this hydrogel system from prior work. For instance, how is the dual-crosslinked structure or its integration with titanium unique compared to existing approaches?

Expand on future potential:

  • Consider including a sentence on prospective in vivo validation or clinical translation, such as: “Future in vivo studies will be needed to confirm biocompatibility and infection-preventing efficacy in physiological environments.”

Typographical cleanup:

  • "temperature/pH sensitive" → "temperature- and pH-sensitive"
  • Ensure consistent formatting of chemical names and abbreviations, e.g., "poly(lactic acid)" instead of "poly (lactic acid)"

Comments on the Quality of English Language

The manuscript would benefit from a thorough English language review. While the scientific content is understandable, there are several grammatical errors, awkward phrasings, and inconsistent sentence structures throughout the text that affect readability. Improving the clarity and flow of the language will enhance the overall quality and professional presentation of the work.

Author Response

(The authors gave the same response as above.)

Reviewer 3 Report

Comments and Suggestions for Authors

Manuscript entitled “Synthesis of Temperature/pH Dual-Responsive Double-Cross-linked Hydrogel on Medical Titanium Alloy Surface” describes synthesis of temperature/pH dual-responsive hydrogels on the surface of medical titanium alloys using Surface-Initiated Atom Transfer Radical Polymerization. MEO2MA and OEGMA were used as temperature-sensitive monomers, DEAEMA as pH-sensitive monomer, PLA as the physical cross-linker, MBA as the chemical cross-linker and modified polydopamine as the initiator. I suggest acceptance after revision.

My comments:

  1. This sentence (in Introduction) is really hard to follow: “To address these limitations, inspired by the structure and lubrication system of natural joints, hydrogel as a drug carrier for topical therapy, researchers have tried to combine hydrogel with artificial bone to form a gradient composite material with a “hard base and soft surface”.” The authors should change it (make it shorter, split in two sentences…).
  2. The author should write the full names for the abbreviations PEG and LCST (In Introduction). The full names are not mentioned in the Introduction.
  3. The authors should consider changing or deleting this sentence (In Introduction): “The pH-sensitive monomer diethylaminoethyl methacrylate (DEAEMA) has a tertiary amine group, which undergoes protonation when the ambient pH is less than pKa, and deprotonation when the ambient pH is greater than pKa.” This sentence is too simplified and gives impression that the authors do not understand protonation/deprotonantion process as a function of pKa and pH. (I think that the authors do understand because they explain it better in the section 2.6. I am just saying how it might look like based on the Introduction.)
  4. There are similar papers in the literature describing the synthesis of hydrogel on titanium alloy surface and using a similar combination of monomers, cross-linkers and initiator. It will be useful to compare results from the literature and the results from this paper (experimental conditions during synthesis and properties of the final product).
  5. Section 2.1. Synthesis of hydrogels: “Three different components of hydrogels were synthesized by varying the content of physical cross-linker to comparatively study the properties of hydrogels. Three different components of hydrogels were synthesized by varying the physical cross-linker content.“ There is no need for both sentences.
  6. Why the 1H NMR spectrum of HEMA-PDLA20 and the FT-IR spectrum of unmodified TC4 were not shown?
  7. Section 4.3.1: The title should be the name of technique. Then, phrase “Macromonomers were measured using a 400 MHz 1H NMR spectrometer” should be replaced with “1H NMR spectrum of sample X was recorded on a 400 MHz 1H NMR spectrometer” or similar. Macromonomers are not an analytical signal that can be measured. The authors should make distinction between analytical technique, instrument, sample, analytical signal… and use proper terminology.
  8. Section 4.3.2: The authors should see my previous comment and make necessary corrections. Not just in this section, in the entire manuscript. I pointed out similar things in comments 9 and 10, but I did not continue with similar comments.
  9. Section 4.3.3: The authors should correct following part: “to measure chemical elements on the surface”.
  10. Section 4.3.4: The authors should finish the sentence. Also, the title should be the name of technique.
  11. Section 4.4: Detailed experimental description is need. For example, different temperatures were not mentioned.
  12. Figure capture 12: Probably “secondary water” should be replaced with “secondary distilled water”.
  13. Section 2.6: “The three gels were immersed in PBS buffer solution at T=37 °C, pH=5.3, pH=7.3, and pH=9.3 for 12 h to reach the equilibrium of swelling.” The authors should specify details about these pH values: Are these the pH values of the buffer solution without gel? Or pH values measured after gels were immersed in the buffer? Did the authors check pH values during the period of 12 h? Also, this can be confusing “pH = 5.3 < 7”. The authors should write this expression more clearly.
  14. The authors should correct reference 28 (names of the authors are written differently comparing to other references).

Author Response

(The authors gave the same response as above.)

Round 2

Reviewer 1 Report

Comments and Suggestions for Authors

The manuscript now is markedly improved. However, for clarification on the data presentation, especially for the drug release results, the authors should acknowledge the limitation that the experiments were conducted with a sample size of one, which precludes statistical analysis. Please make such a statement in elsewhere in the manuscript.

Author Response

Dear reviewer,

Thanks for your suggestion. The reply can be found in the attachment.

Reviewer 3 Report

Comments and Suggestions for Authors

I do not think that names "NMR Testing, FT-IR Testing, ..." are appropriate in this case. I suggest these names of techniques: Proton nuclear magnetic resonance spectroscopy (1H NMR),  Fourier-transform infrared spectroscopy (FT-IR), X-ray photoelectron spectroscopy (XPS), X-ray diffraction (XRD). 

Author Response

Dear reviewer, 

Thank you for your suggestion. The reply can be found in the attachment.
